# SOX9 plays an essential role in myofibroblast driven hepatic granuloma integrity and parenchymal repair during schistosomiasis-induced liver damage

Kim Su[1☯], Elliot Jokl[1☯], Alice Costain[1,2☯], Kara Simpson[1], Antonn Cheeseman[1], Alexander Phythian-Adams[1], Kevin N. Couper[1], Andrew S. MacDonald[1,2,3], Karen Piper Hanley[1*]

1 Faculty of Biology, Medicine & Health, Manchester Academic Health Science Centre, University of Manchester, Oxford, Manchester, United Kingdom, 2 Lydia Becker Institute of Immunology and Inflammation, University of Manchester, Oxford, Manchester, United Kingdom, 3 Institute of Immunology and Infection Research, University of Edinburgh, Edinburgh, United Kingdom

☯ Authors contributed equally to the production of this manuscript
* karen.piperhanley@manchester.ac.uk

## Abstract

Schistosomiasis is a neglected, and potentially lethal, parasitic disease that affects hundreds of millions of people worldwide. As part of the schistosome lifecycle, parasite eggs accumulate within the liver where they evoke intense granulomatous pathology, typified by a dense extracellular matrix (ECM) barrier, which serves to contain toxic egg secretions. In severe cases, this progressive and irreversible egg-evoked ECM deposition can lead to pathological scarring, impaired liver function and lethality. Thus, identifying the core regulators that govern ECM deposition may aid discovery of new therapeutic targets for schistosomiasis. The transcription factor Sex determining region Y-box 9 (Sox9) is a known regulator of pathological scaring. We found that, following *Schistosoma mansoni* infection, SOX9 was ectopically expressed in myofibroblasts within the granuloma and in surrounding hepatocytes. In the absence of SOX9, granuloma size was significantly diminished, and mice failed to produce a robust ECM barrier around eggs, resulting in more diffuse liver injury and scattered distribution of immune cells. Immunologically, SOX9 loss in both naïve and infected mice led to an increase in hepatic neutrophil and monocyte proportions, with the expansion of Ly6c[lo] monocyte populations in infected SOX9 deficient mice only. Infected SOX9–deficient mice also displayed exaggerated Type 2 inflammation, including pronounced eosinophilia. These data highlight the importance of SOX9 for intact hepatic granuloma formation during schistosomiasis and suggest SOX9 or its related factors may provide attractive future targets for meeting the clinical need to limit and/or reverse fibrotic disease.

which permits unrestricted use, distribution, and reproduction in any medium, provided the original author and source are credited.

**Data availability statement:** All relevant data are within the manuscript and its Supporting Information files.

**Funding:** This work was supported by the Medical Research Council (MR/P023541/1 to KPH). AM was supported by funding from the Lydia Becker Institute and the MRC (MR/W018748/1). We acknowledge support from the Wellcome Trust (105610) and Biotechnology and Biological Sciences Research Council (BB/S019324/1 to KNC). KPH is a member of the Wellcome Trust supported Centre for Cell-Matrix Research (203128/Z/16/Z). KS was supported by the North West MRC Fellowship MR/N025989/1 in Clinical Pharmacology & Therapeutics. The funders had no role in study design, data collection and analysis, decision to publish, or preparation of the manuscript.

**Competing interests:** The authors have declared that no competing interests exist.

## Author summary

Mammalian infection with schistosome worms results in the deposition of parasite eggs in the liver, where they secrete organ damaging toxins. In response, the liver generates a cellular granuloma barrier rich in extracellular matrix to limit these secretions and protect the overall organ. As in other liver injuries, SOX9 becomes progressively expressed in multiple cell types during the time course of schistosome infection. To understand the role of SOX9 in the liver response to schistosomes we utilised a global SOX9 deficient mouse model. These mice show reduced and disorganised granuloma formation during schistosome infection, with disrupted hepatic immune profiles. This suggests that SOX9 is required to form a robust and coordinated granuloma barrier that limits liver damage in this important but neglected parasitic disease.

## Introduction

Schistosomiasis is a neglected tropical disease (NTD) caused by parasitic blood flukes of the genus *Schistosoma*. An estimated 236.6 million people worldwide received preventative treatment for schistosomiasis in 2019 and, as an NTD, it has been ranked second in terms of years lived with disability (YLD) in the Global Burden of Disease 2010 Report [1,2]. In humans the most common schistosomiasis species affecting the liver is the intestinal species, *S. mansoni*. As part of *S. mansoni* infections, adult worms live and lay their eggs within the mesenteric vasculature. These eggs either exit their mammalian host by traversing the intestinal wall or become trapped within the liver or other distal organs. Unlike adult worms which have evolved evasive strategies to avoid immune detection [3] schistosome eggs are highly visible to the host immune system, where their tissue entrapment elicits a strong inflammatory response, generating cellular structures known as granulomas, which are central to schistosomiasis-associated pathology [4]. Hepatic granuloma formation is characterised by an influx of immune cells, often type 2 in nature, alongside accumulation of tissue damaging and fibrotic extracellular matrix (ECM) from activated hepatic stellate cells (HSCs, liver specific myofibroblasts). ECM deposition is thought to be beneficial to the host as it limits the diffusion of egg secretions, minimising parenchymal inflammation and damage [5]. However, in severe cases granuloma driven fibrosis progresses, resulting in portal hypertension and, ultimately, liver failure. Treatment and killing of schistosome worms is possible but this does not reverse severe fibrosis or alleviate morbidity associated with chronic infection. There is an urgent clinical need to develop new treatments targeting factors that regulate liver fibrosis in patients with chronic *S. mansoni*.

Sex-determining region Y-box 9 (SOX9) is a master transcription factor that plays a critical role in various developmental processes, including the coordination of ECM during bone formation [6]. We have previously identified SOX9 as a core factor that becomes expressed during activation of HSCs where it is responsible for production

of multiple fibrotic ECM components. Loss of SOX9 *in vivo* alleviates carbon tetrachloride (CCl4) and bile duct ligated (BDL) induced liver fibrosis, including a reduction in collagen deposition and HSC activation, and improved liver functionality [7–9]. We therefore hypothesised that loss of SOX9 might also lessen the fibrosis observed in chronic *S. mansoni* infection and identify new targets for future therapeutic development in chronic *S. mansoni* patients.

In this study, we utilised an inducible global SOX9 deficient mouse model to study the downstream events and granuloma fibrosis following *S. mansoni* infection in the presence and absence of SOX9. Our data show that SOX9 is expressed progressively during infection in multiple liver cell types, most extensively in injured hepatocytes adjacent to scarring, highlighting additional cell specific roles identified for SOX9 in the context of injury and parenchymal repair [10,11]. Although granuloma fibrosis was indeed reduced in the absence of SOX9, failure to form a robust ECM barrier around the egg resulted in a more widespread and diffuse myofibroblast activation and "micro-fibrosis". Coinciding with these pathological alterations, SOX9 loss dramatically altered hepatic immune cell responses, including more pronounced eosinophilia, a reduction in CD4+ T cell proportions and heightened frequency of Ly6lo monocytes. Taken together, these data highlight the critical role for SOX9 in balancing both the coordinated recruitment of immune cells to the site of injury for robust granuloma barrier formation and in supporting parenchymal repair from uncontained immune insult. Alongside our previous studies, this suggests a pro-fibrotic role for SOX9 across injury immunotypes and indicates factors downstream of SOX9 may yield therapeutic targets against fibrosis and to promote corrective repair during schistosomiasis and in other fibrotic diseases.

## Results

### SOX9 is progressively expressed in multiple hepatic cell types during S. mansoni infection

During schistosomiasis, hepatic granulomas vary in size, structure and collagen content, influenced by factors such as egg maturity, location of egg deposition and the time point of infection[12,13]. Over the course of murine infection, the liver becomes overwhelmed by extensive granulomatous inflammation (S1A Fig). We first sought to localise SOX9 expression during *S. mansoni* infection. (Fig 1). In naïve livers robust SOX9 expression is limited to cholangiocytes, with weak expression in a small subset of hepatocytes. Alongside the myofibroblast marker alpha Smooth muscle actin (αSMA) and picrosirius red (PSR; for collagen deposition), SOX9 was progressively increased as fibrosis became established in the course of infection, reaching significance by day 56 (Fig 1A–D ). During infection and consistent with other liver injury models, SOX9 expression occurred within activated HSCs in the granuloma scar, cholangiocytes demonstrating ductal hyperplasia, and injured hepatocytes as determined by co-localisation with the hepatocyte specific marker HNF4α (Fig 2). The proportion of SOX9+ cells was greater in the immediate periphery of the granuloma than within the granuloma itself, suggesting a greater contribution of cholangiocytes and damaged hepatocytes to overall SOX9 expression (S1B,C Fig).

### Loss of Sox9 prevents vigorous fibrotic granuloma formation

With evidence of SOX9 activity in multiple cell types, we used tamoxifen-inducible global deficiency (RosaCreER; Sox9fl/fl) to eliminate SOX9 expression prior to *S. mansoni* infection (S2A Fig). Long-term SOX9 deficiency was associated with increased mortality, reflecting the detrimental effect of global SOX9 loss (S2B Fig). An infection length of 51 days was determined as a balance between animal welfare considerations and likelihood of observing a robust phenotype based on the known time course of SOX9 expression during infection.

On completion of the study, infected control and SOX9-/- groups displayed significant hepatomegaly (**Fig 3A**) compared to their respective naïve groups, reflecting both parasitic burden and fibrosis. Infected SOX9-/- mice also showed increased splenomegaly compared to both SOX9-/- naïve and control infected mice (**Fig 3B**). This is an indicator of portal hypertension, and subsequent engorgement of the spleen with blood, suggesting enhanced portal shunting or ectopic egg spread. SOX9-/- mice showed a significant reduction in granuloma size as a proportion of liver area (**Fig 3C**), suggesting a failure to robustly produce a fibrotic barrier between the egg and the surrounding parenchyma. Interestingly, KO mice showed

PLOS Pathogens

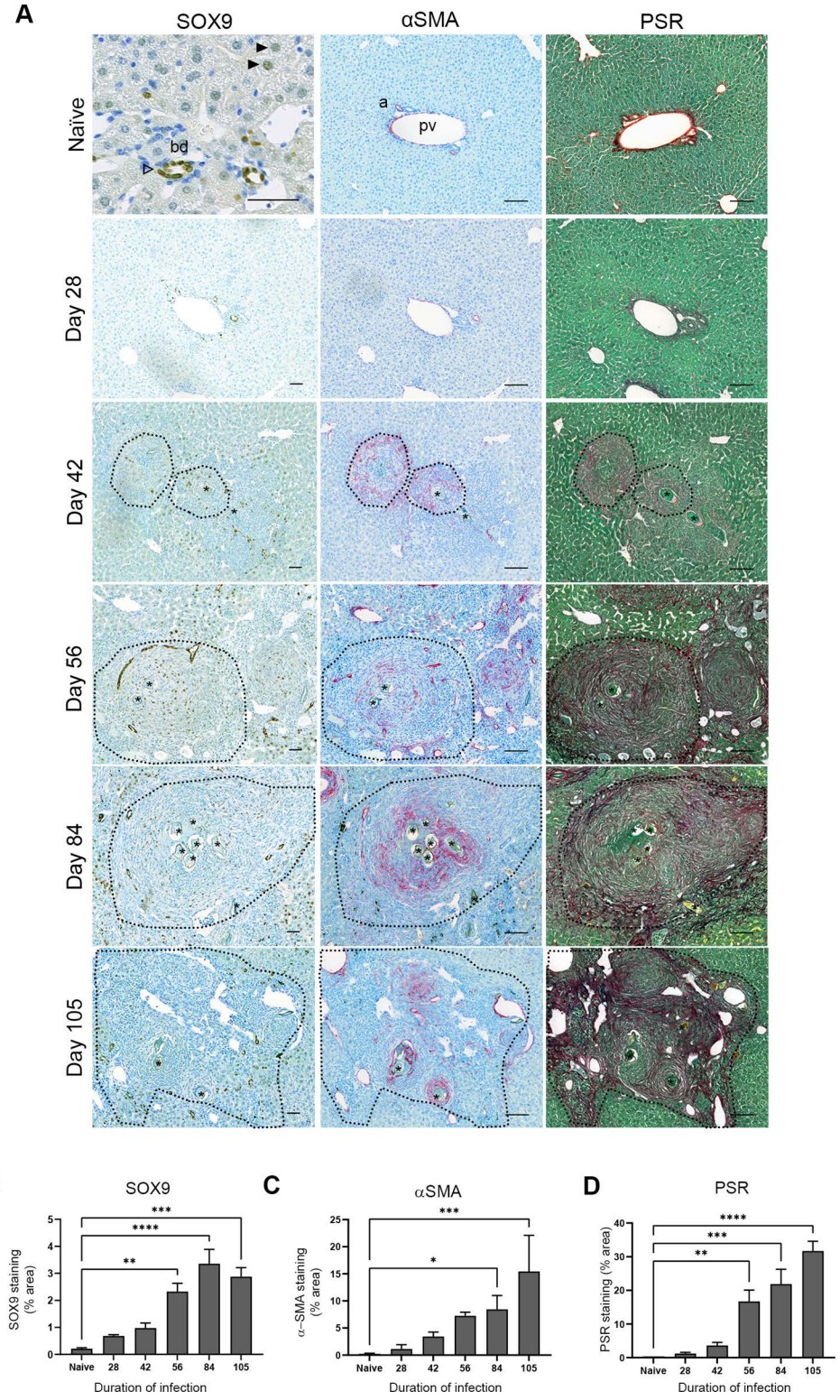

**Fig 1. SOX9 is expressed in areas of HSC activation and fibrosis deposition in *S. mansoni* infection.** (A) Tissue sections from naïve and *S. mansoni* livers stained for SOX9 (brown), α-SMA (magenta), and PSR (red, counterstained with fast green). In naïve mice SOX9 is detected in cholangiocytes (open arrowhead) and occasional hepatocytes (black arrowheads). A-SMA and PSR expression localise to vascular regions. In infected mice

SOX9 expression is upregulated and localises in part to areas of α-SMA and PSR staining as progressive fibrosis develops in the granuloma (dotted black line). All scale bars 50μm. bd = bile duct, a = artery, pv = portal vein, * = granuloma egg. (B) SOX9 IHC quantification, (C). α-SMA IHC quantification and (D) PSR IHC quantification throughout the duration of *S. mansoni* infection. Data obtained by using 10 images per X20 field. 1 lobe per animal and n = 3 animals per time point. Statistical significance determined by one-way ANOVA and Dunnett's multiple comparisons test (*p < 0.05, **p < 0.01, ***p < 0.001, ****p < 0.0001).

a significant reduction in hepatic, but not small intestinal egg burden compared to controls (S2C and S2D Fig). This may reflect differences in liver architecture, with eggs being redirected to extrahepatic sites due to impaired ECM deposition and fibrosis.

Significantly, SOX9-/- mice showed a reduction in granuloma PSR staining and tissue-level COL1 expression by western blot. However, PSR and αSMA localisation showed a more diffuse pattern of expression, being more widely dispersed throughout the tissue (Figs 3D and S3). This is consistent with a "leaky granuloma", where a failure to produce an effective barrier around the egg permits the leakage of toxins and other factors which cause damage throughout the tissue [5]. However, no significant differences were observed in levels of necrosis between the SOX9-/- and control infected groups (S4 Fig).

### SOX9 loss restructures hepatic immunity

We next looked to explore the immunological consequences of SOX9 deficiency during *S. mansoni* infection. (Fig 4). As expected [12], irrespective of SOX9 depletion status, schistosome infections were accompanied by significant eosinophilia, but with SOX9 -/- mice showing a strong tendency towards higher eosinophil proportions than WT infected controls (Fig 4A). Interestingly, although total monocyte proportions remained unaltered between naïve and infected mice (Fig 4A), we observed significant expansion of 'restorative' Ly6C$^{lo}$ monocytes in infected SOX9-/- mice, coincident with a decrease in 'proinflammatory' Ly6C$^{hi}$ monocytes. These monocyte alterations were not visible in naïve SOX9-/- animals. Neutrophilia was also observed in both naïve and infected SOX9-/- mice relative to controls, with significance reached in infected SOX9 deficient mice only (Fig 4A).

CD4+ T cells have an essential role in granuloma orchestration [14,15]. Along with reduced granulomatous pathology (Fig 3), we observed a significant reduction in the proportion of CD4+ T cells in the livers of infected SOX9-/- mice (Fig 4B). Further analysis of these CD4+ T cells showed elevated expression of the IL-33 (alarmin) receptor ST2, alongside CD25, a marker linked to T cell functionality and regulation (Fig 4B). A further hint towards increased expression of the Th17 transcription factor, RORγt was also shown (Fig 4C). In the absence of SOX9, CD4+ T cells from infected animals also demonstrated a higher propensity to secrete IL-17 than WT animals, but with no differences in IL-4, IL-5, IL-10 or IFNγ production (Fig 4D).

### SOX9 loss alters the spatial organisation of hepatic ECM and immune cell populations

We then examined whether failure to form structured granulomas led to altered spatial organisation of immune cell populations in the liver. SOX9 loss was shown to impact the distribution of F4/80+ cells (likely macrophages) by IHC. In WT mice F4/80+ cells were largely confined to granuloma regions, whilst in SOX9-/- mice, F4/80+ cells were more randomly dispersed across the tissue (Figs 5A and S5A). Furthermore, similar patterns were observed when inspecting the distribution of Ym1+ cells, Ym1 being a marker for wound healing, alternatively activated (AA) macrophages (S5B Fig). Whilst Ym1 staining was not visible in naive mice, we observed organised clustering of Ym1+ cells around schistosome eggs in the livers of infected SOX9 intact animals. This pattern was dramatically fragmented by the depletion of SOX9, where Ym1+ cells showed scattered distribution across the liver parenchyma.

Hyperion Imaging Mass Cytometry was used to further characterise the fibrotic consequences of SOX9 deficiency (Figs 5 and S6). As expected, naïve mice displayed minimal ECM deposition, with diffuse collagen IV (COL4) across

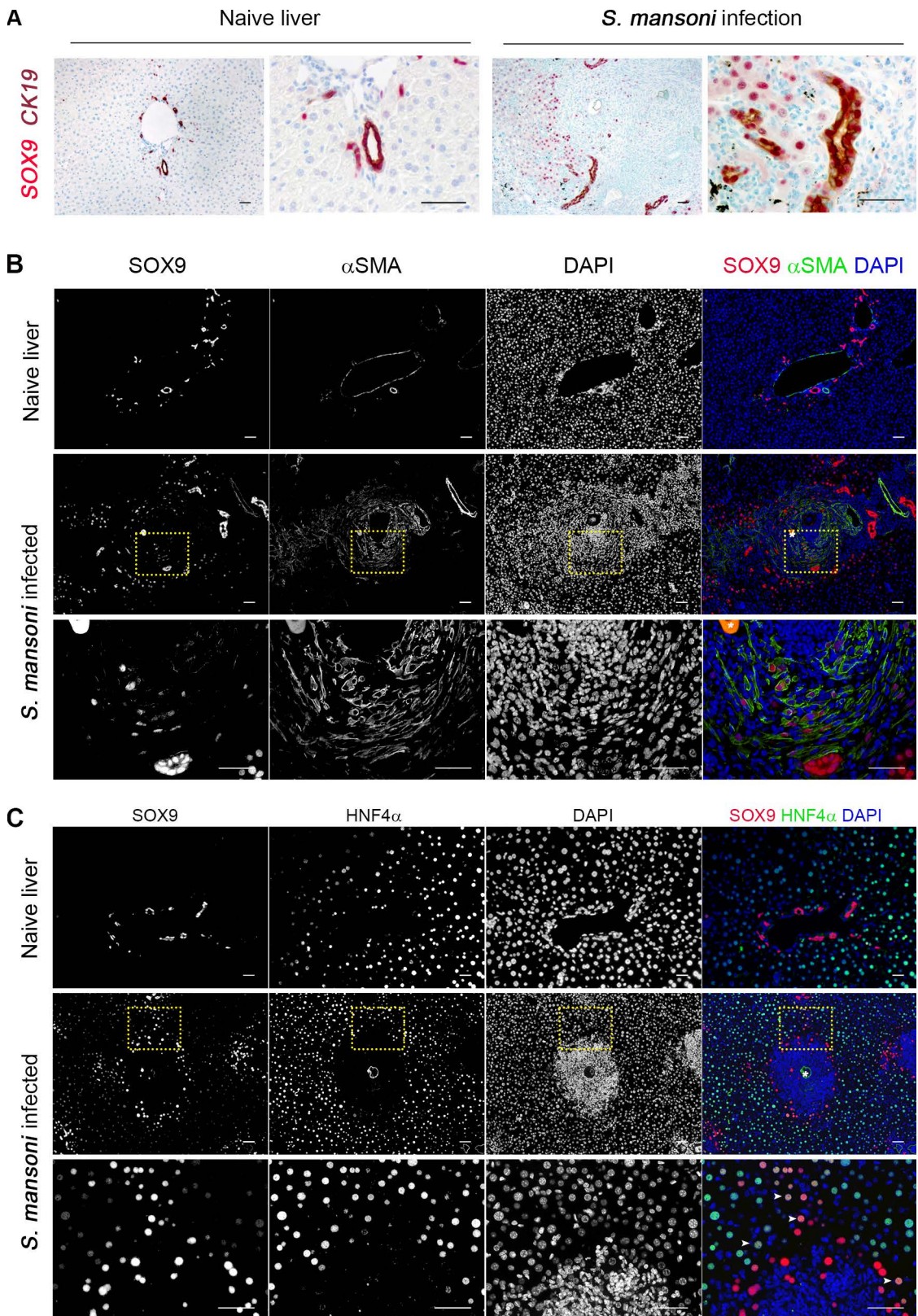

**Fig 2. SOX9 is expressed in three distinct cell types of the liver in *S. mansoni* infection.** (A) Naïve and *S. mansoni* infected livers underwent dual IHC for SOX9 (magenta) and CK19 (red) with toluidine blue as counterstain. SOX9 localises to CK19⁺ cholangiocytes in the naïve liver. In *S. mansoni*

infection, SOX9 is detected in CK19+ ductular reaction cells. All scale bars 50μm. (B) Dual IF for SOX9 (red) and α-SMA (green) in naïve and *S. mansoni* infected livers, with DAPI (blue) as nuclear counterstain. SOX9 is expressed in nuclei of α-SMA+ cells around the schistosome egg (white asterisk), reflecting upregulation of SOX9 in activated HSCs. All scale bars 50μm. Bottom row is a zoom of the region demarcated by the dotted yellow box. (C) Dual IF for SOX9 (red) and HNF4α (green) in naïve and *S. mansoni* infected livers, with DAPI (blue) as nuclear counterstain. Dual expression of SOX9 and HNF4α+ hepatocytes are seen at margins of granulomas (white arrows). All scale bars 50μm. Bottom row is a zoom of the region demarcated by the dotted yellow box.

the liver, and collagen I (COL1) staining exclusive to the circumference of hepatic vasculature. No differences were observed between SOX9-/- and WT naïve mice. In the egg-free regions of infected livers, ECM patterns were similar to naïve animals, with enhanced frequencies of proliferating Ki67+ cells and additional clustering of macrophages positive for the calcium binding protein Iba1, which is linked with macrophage activation and scenarios of inflammation [16]. In egg regions, granulomatous inflammation was more robust and defined in WT mice than SOX9-/- mice. WT granulomas were typified by a dense and concentric network of COL1 expression, extending from the innermost to outermost regions of the granuloma. COL1 distribution was interlaced with fragments of heparin sulphate and densely interwoven with fibronectin, both of which were more heavily localised within the inner section of granuloma. In stark contrast, SOX9-/- granulomas lost their coordinated halo of collagen, instead presenting a more disorganised and diffuse arrangement of ECM components, and the staining for these matrix components was less intense, reminiscent of reduced fibrosis. WT granulomas were decorated by a peripheral ring of proliferative cells. This pattern was lost in SOX9-/- mice, suggesting a wider, less spatially controlled distribution of cell proliferation, potentially relevant to corrective repair.

## Discussion

Intact granuloma formation plays a critical role in limiting liver damage following entrapment of schistosome eggs in the liver. Although the anti-helminthic drug praziquantel can eliminate adult schistosome worms, in chronic infection the residual fibrotic plaque remaining after granulomatous resolution can result in chronic liver disease. In this study we used *in vivo* models to interrogate core mechanisms controlling ECM coordination and resolution to identify much needed avenues for innovative anti-fibrotic therapies. Our work focussed on the core ECM regulator, SOX9, known to be induced in activated of HSCs [8,17]. Similar to our previous work on liver disease [7,9], following *S. mansoni* infection SOX9 was ectopically expressed in activated HSCs, contributing to ECM deposition in the granuloma and hepatocytes surrounding the granuloma. The localisation of SOX9 in hepatocytes is reminiscent of our previous work identifying SOX9 positive hepatocytes aligning the scar interface during progressive liver disease, with similarities to the ductal plate during liver development [7]. However, in human liver disease, increasing levels of SOX9 in hepatocyte populations is a prognostic and diagnostic measure that parallels disease severity [9]. Although SOX9 plays a regulated role in the regenerative response of the liver in acute damage, ectopic hepatocyte expression in chronic liver disease is thought to be associated with an impaired regenerative/ disease response [11].

In keeping with its role in myofibroblast activation and ECM deposition, the absence of SOX9 reduced fibrotic granuloma and myofibroblast activation within the liver during *S. mansoni* infection. Critically, loss of SOX9 and subsequent depletion of intact granuloma ECM had a significant impact on recruitment, maintenance and functionality of immune cell distribution within the liver. In particular, SOX9 deficiency led to the dispersed arrangement of F4/80+ cells and altered CD4+ T cell phenotypes, with pronounced eosinophilia and monocyte recruitment; potentially suggesting signals from the fibrotic ECM help tether immune cells to the granuloma or modulate their function. CD4+ T cells have an established role in the orchestration of intact granuloma formation during schistosomiasis [14,15,18,19], where a precise balance between Th1 and Th2 cells, and regulatory cells (including Regulatory T cells [Tregs] and B cells [Bregs]) is needed to avoid severe and potentially life-threatening pathology [20]. It is possible that ECM components, secreted by SOX9 expressing HSCs,

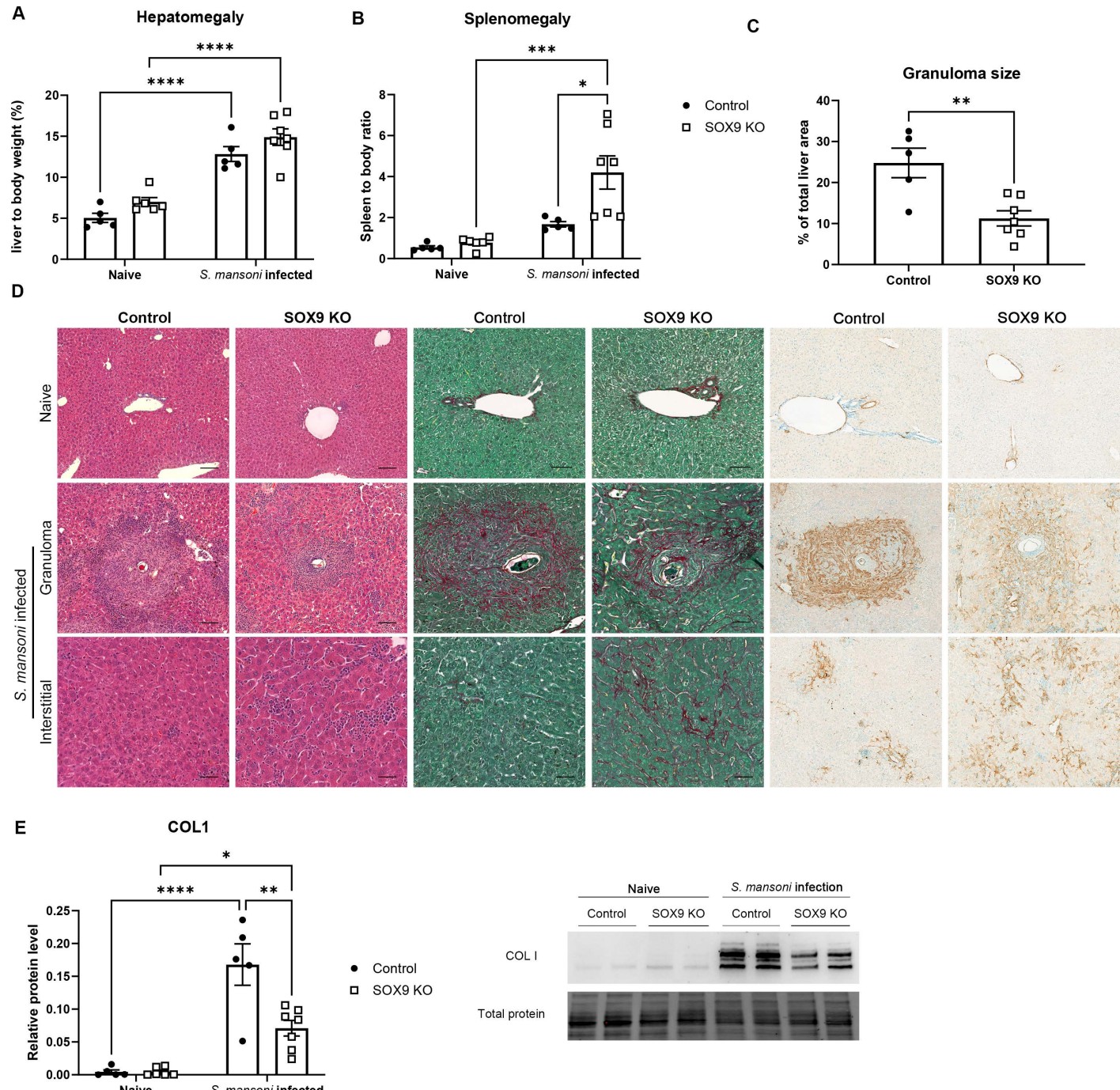

**Fig 3. Failure to generate robust granuloma results in interstitial fibrosis in SOX9KO.** (A) Spleen weight and (B) Liver weight as a proportion of body weight. (C) Area of granuloma as a proportion of total liver area. (D) Representative H&E, PSR and aSMA IHC of naïve tissue, granulomas of infected mice and interstitial tissue between granulomas. SOX9KO granulomas are smaller and less well defined. SOX9KO mice have increased interstitial fibrosis. (E) Significantly reduced COL1 protein in infected SOX9KO than WT controls at the tissue level as detected by WB. Statistical analysis using (A, B, E) Two-way ANOVA with Tukey's multiple comparisons and (C) unpaired Student's t-test. *p < 0.05, **p < 0.01, ***p < 0.001, ****p < 0.0001. n = 5–7 animals per group.

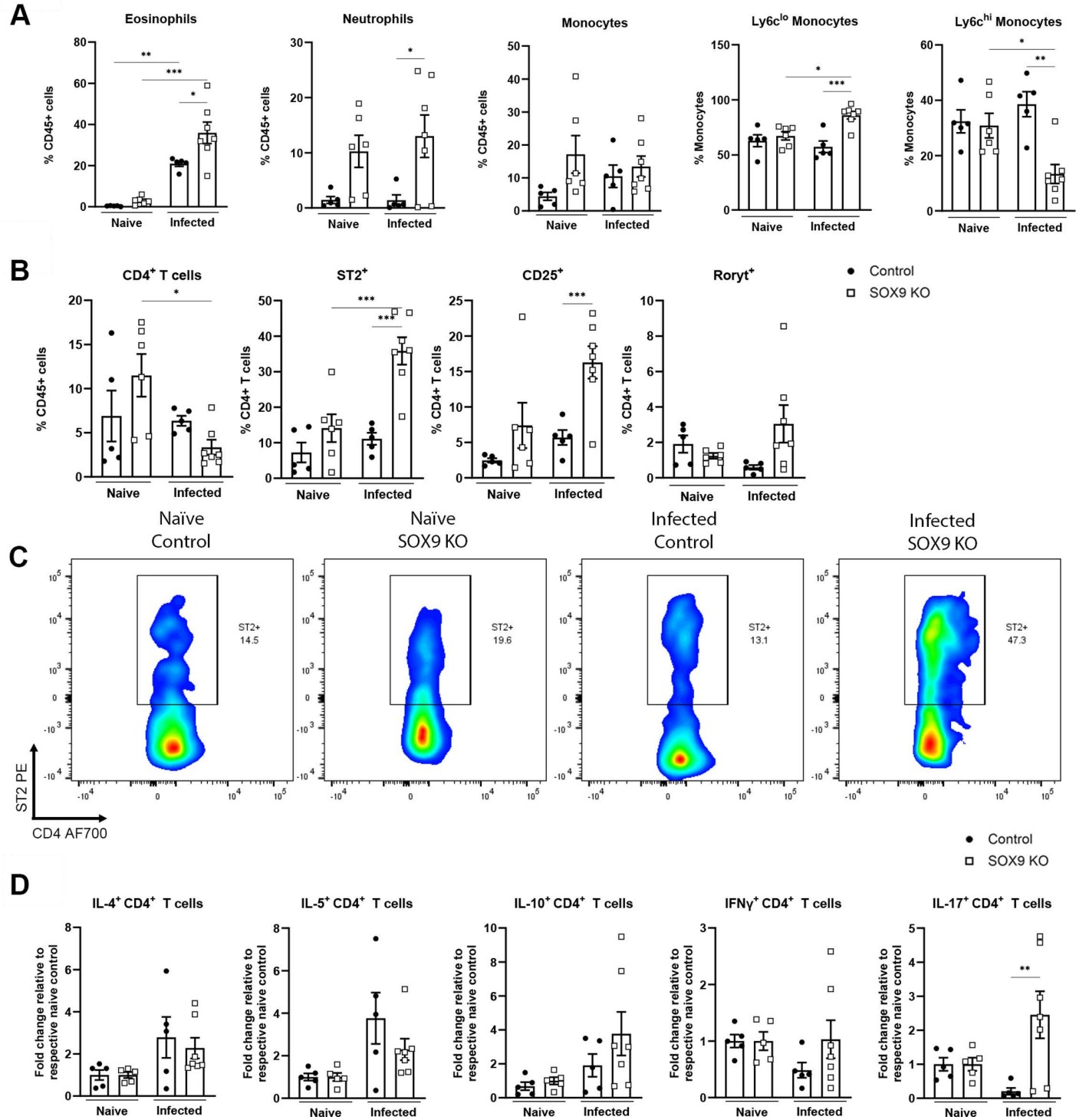

**Fig 4. Sox9 depletion reconfigures hepatic immune profiles during S. mansoni infection.** The frequency of indicated myeloid cell populations (A) total CD4+ T cells and (B) their expression of ST2, CD25 and RORγt in the livers of naïve and infected, Cre-ve or RosaCre Sox9 KO mice. (C) Representative flow plots for ST2+ gating, pre-gating on liver CD45+TCRβ+CD4+ cells. (D) Cytokine secretion in PMA-stimulated hepatic CD4+ T cells. Data presented as mean +/- SEM. Data shown are from 2 pooled experiments. N = 1-4 per experimental group per experiment. Significant differences were determined by one-way ANOVA with Tukey's post-hoc testing. * p < 0.05, ** p < 0.01, *** p < 0.001.

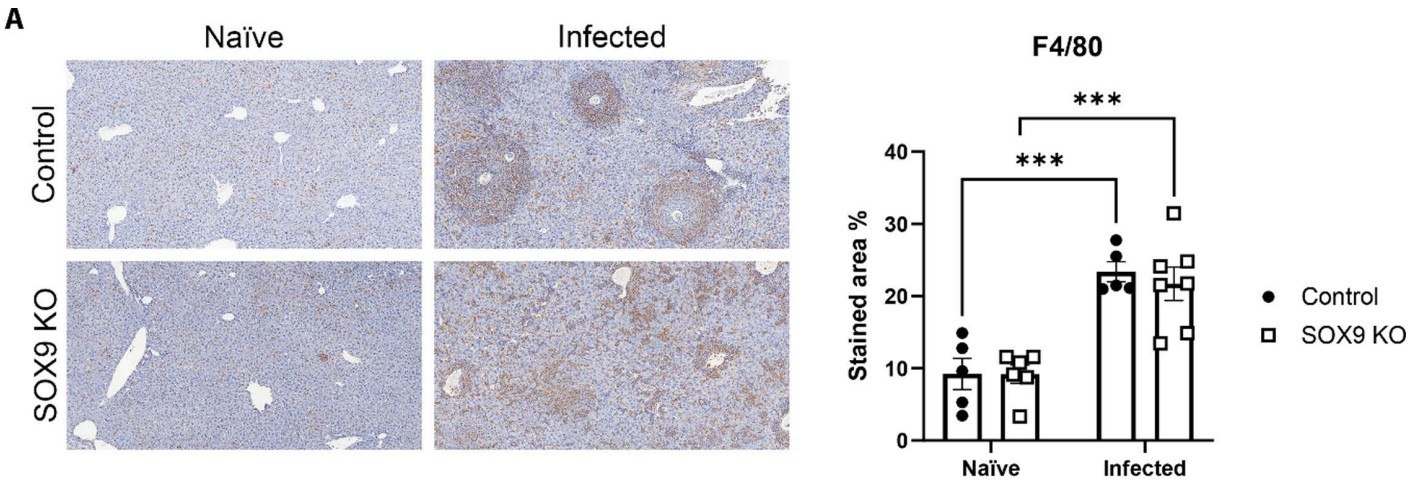

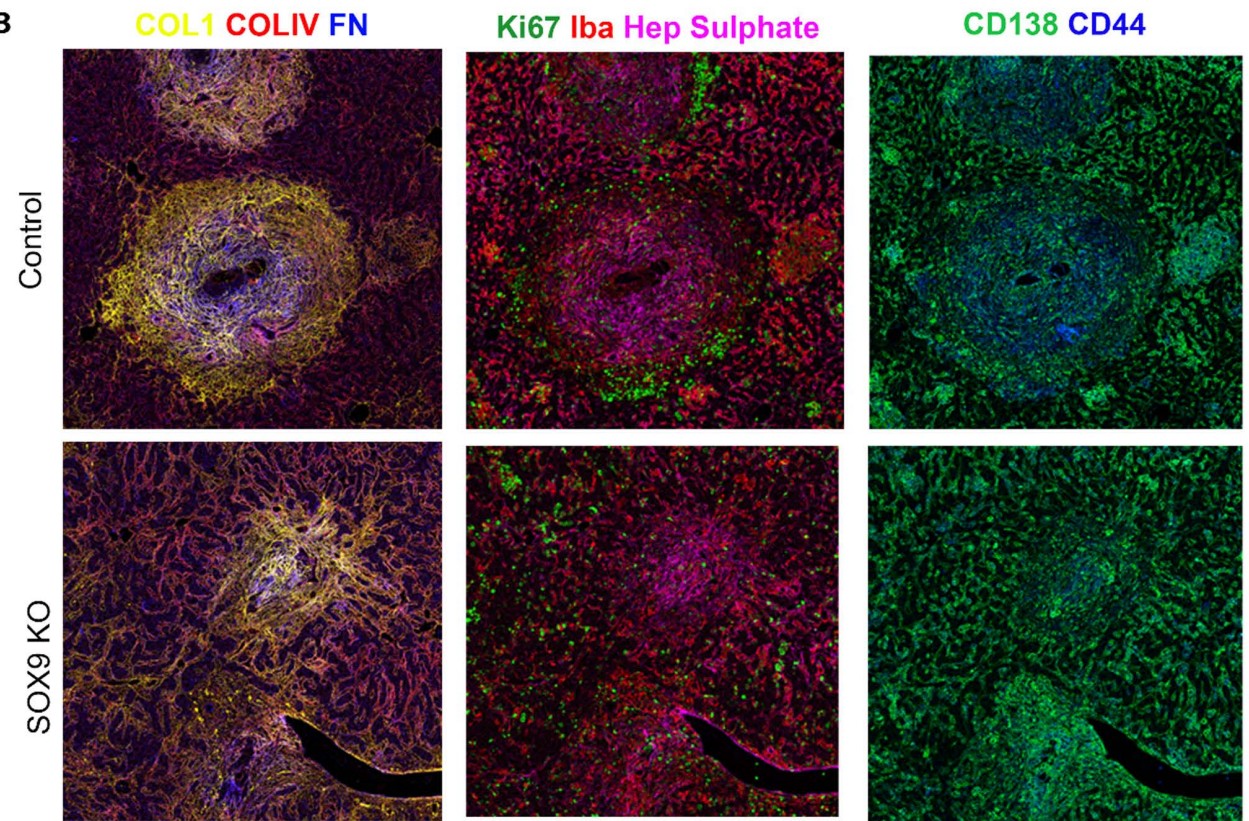

**Fig 5. Aberrant spatial organisation of immune cells in SOX9 deficient animals.** (A) F4/80 IHC staining of infected control and SOX9KO animals and quantification of stained area as a proportion of tissue area (n = 5–7 animals per group). Significant differences were determined by one-way ANOVA with Tukey's post-hoc testing. *** = p < 0.001. (B) Representative mass cytometry images of liver from infected control and SOX9KO animals. Staining targets are identified by the colour key.

are needed to anchor cell types like CD4[+] T cells and Ym1[+] cell to the granuloma, and thus in the absence of SOX9, there is a deficit their recruitment and/or maintenance within the granuloma leading to more diffuse damage. Indeed, liver pathology is known to be more severe in settings of Th2 deficiency [15]. Alternatively, the observed alterations in hepatic immunity could represent a compensatory mechanism that is triggered in response to enhanced parenchymal damage, in an attempt by the host to alleviate egg evoked tissue damage. Indeed, infected SOX9[-/-] mice display a higher proportion of Ly6c[lo] monocytes, where these populations have previously demonstrated restorative functions in inflammatory liver conditions, including rotavirus mediated liver inflammation [21] and chronic $CCl_4$-induced liver fibrosis [22]. The damage elicited by eggs and/or unshackled egg toxins could provoke the immediate release of the alarmin IL-33, from immune and non-immune cells, resulting in the enhanced expression of its receptor ST2 on CD4[+] T cells [23]. It would be valuable to explore whether IL-33 is influences the flavour of host immunity in this model, given recent evidence showing the source of IL-33 significantly shapes immune responses: epithelial-derived IL-33 promotes type 2 immunity, while antigen-presenting cell-derived IL-33 drives regulatory responses [24].

Steady state turnover of extracellular matrix in the naïve liver may also be important for immune regulatory functions of the liver. In particular, the lack of SOX9 induction in hepatocytes during an acute injury response may have profound implications for homeostatic repair mechanisms [10,11,25]. Interestingly, despite the lack of SOX9, we still observed microfibrosis in the liver during schistosome infection, suggesting other regulatory transcription factors (and likely cell types) are capable of at least some ECM deposition in this context. These questions could be explored in future work using lineage specific SOX9 deficiency to systematically assess the relative contribution of different cell types to the phenotypes we observe in the globally deficient mice during *S. mansion* infection. For example, eliminating SOX9 expression in hepatocytes while maintaining SOX9 in HSCs would allow specific focus on how an altered hepatocyte injury response influences hepatic immune modulation and fibrosis during infection.

Taken together, our data reveal an important role for SOX9 in the maintenance of intact hepatic granulomas and in the corrective reparative response during schistosome infection. In the ongoing search for anti-fibrotic therapies, targeting SOX9 may be successful in treating fibrosis associated with schistosomiasis and other diverse aetiologies. However, a greater understanding of the pathophysiological response of immune cell activation, particularly type 2 polarisation, in broad liver disease would be a valuable step in designing approaches to combat liver fibrosis more generally.

## Materials and methods

### Ethics statement

Animal experiments were performed under a project license granted by the UK Home Office (PPL: PP8444846) following ethical review and approval by the University of Manchester Animal Welfare Ethical Review Board or University of Edinburgh and performed in accordance with the United Kingdom Animals (Scientific Procedures) Act of 1986.

### Mice

Experiments were performed using female C57BL/6 or Sox9[fl/fl];RosaCreER mice [7], with experiments initiating at 8–10wks of age.

### *Schistosoma mansoni* infection and Sox9 depletion

Biomphalaria glabrata snails infected with S. *mansoni* parasites were obtained from Prof Karl Hoffman at Aberystwyth University. Mice were percutaneously infected with 40 S. *mansoni* cercariae, with infections lasting 28, 42, 56, 84 and 105 days in duration for the experiments assessing SOX9 expression during the time course of infection, and 51 days in duration for the Sox9[fl/fl];RosaCreER model. For quantification of parasite burden, S. *mansoni* eggs were isolated from the caudate lobe (approx. 0.2g in weight) or ileum and digested overnight in 5% KOH. To induce conditional depletion of Sox9, Sox9[fl/fl];RosaCreER mice were intraperitoneally injected with tamoxifen at doses of 100mg/kg (days 1, 2 and 4) or

60mg/kg (days 15, 29, 43 and 57). Mice were infected with schistosoma parasites 8 days post the first tamoxifen injection and culled on day 51 of infection (See S1 Fig). Genotypes were validated by PCR as described in [7].

### Cell isolation

Single-cell suspensions of liver were prepared for subsequent flow cytometry. Livers were perfused, minced with scalpel blades, and incubated at 37°C for 20 minutes with 1mg/ml collagenase IV (Roche) and 10ug/ml DNAse (Sigma) in RPMI-1640 supplemented with 50 U/ml penicillin and 50 µg/ml streptomycin (Invitrogen), 20mM Hepes (Sigma) and 2mM L-Glutamine (Sigma). Samples were subsequently placed on ice and topped with complete RPMI-1640 media containing 3% FCS (sigma) to stop the reaction. Digested livers were passed through a 70 µm filter and pelleted and centrifuged at 500 g. Pelleted cells were passed through a 40 µm cell strainer (for removal of S. mansoni eggs), followed by RBC lysis, resuspension in PBS supplemented with 2% FBS, 2 mM EDTA. For intracellular cytokine staining, cells were plated at $0.4\text{-}1 \times 10^6$ cells/well in volumes of 200µl and stimulated for 3h in the presence of 30ng/ml PMA, 1 µg/ml Ionomycin and 1 µg/ml Golg Stop (BD), before processing for flow cytometry.

### Flow cytometry

Single cell suspensions were washed with PBS, stained for viability with ZombieUV (1:2000; Biolegend) for 10 min at room temperature. Cells were stained for surface markers for 30 min at 4°C with antibody cocktails created in FACS buffer (PBS containing 2% FBS and 2 mM EDTA) containing 5 µg/ml FC block (αCD16/CD32; 2.4G2; Biolegend). For intracellular markers, cells were permeabilized with BD Cytofix/Cytoperm Fixation/Permeabilization Solution for 1 h at 4°C, before staining with relevant intracellular antibodies. Samples were analysed by flow cytometry (LSR Fortessa, BD) and data analysed using FlowJo V10 software. S1 Table provides a list of antibodies used.

### Immunohistochemistry and immunofluorescence

For histological imaging, pieces from three independent liver lobes per mouse were fixed in 10% NBF overnight at 4 degrees, dehydrated through a series of graded alcohols and embedded in paraffin blocks. Tissues were subsequentially cut into 5µm sections (Leica RM2235 microtome). H&E, Picro-Sirius Red and immunohistochemistry staining were performed as previously described. Primary antibodies used were Anti-SOX9 (Millipore; AB5535; 1:2000), Anti-aSMA (CST; 19245; 1:300), Anti-CK19 (DSHB; TROMA-III; 1:300), Anti-HNF4a (R&D; H1415-00; 1:400) and Anti-F4/80 (Abcam; ab6640; 1:200).

For Ym1 immunofluorescence, sections were soaked in xylene, passed through graded alcohols, and then treated with Retrievagen A (BD) antigen retrieval solution in a microwave at 95 degrees for 20 minutes. Sections were left to cool for 30 minutes, before two washes in PBS, then permeabilised by incubation with 0.5% Triton X100 (PBS) for 20 minutes. After two subsequent wash steps (PBS) sections were blocked with blocking medium (2% normal donkey serum, 0.05% Tween, 1% BSA in PBS) for 30 minutes. Sections were washed twice more (PBS) before blocking endogenous biotin activity (Invitrogen) according to manufacturer's instructions. Samples were incubated overnight at 4°C with biotinylated Anti-Ym1 (R&D systems; 1:50; BAF2446) in 1% BSA in PBS. Following two washes in PBS, samples were incubated for 1 h at RT with NorthernLights Streptavidin NL637 secondary antibody (R&D Systems; 1:200; NL998). Sections were washed twice, stained with Dapi per the manufacturer and cover slipped with Fluoromount Aqueous Mounting Medium (Sigma). Images were collected on a Zeiss Axioimager.D2 upright microscope using a 10x Plan-neofluar objective and captured using a Coolsnap HQ2 camera through Micromanager software v1.4.23. Fiji ImageJ was used for all image analyses, except for granuloma/peri-granuloma quantification of SOX9. In QuPath [26] (v0.51), the positive cell detection tool was used to identify the proportion of SOX9+ cells within manually annotated granulomas, and a 50µm border around the granuloma edge which we defined as the peri-granuloma region.

## Hyperion imaging mass cytometry

Antibodies for imaging mass cytometry were purchased pre-conjugated from Standard BioTools (formerly Fluidigm) or purchased unconjugated and BSA-free from commercial suppliers and conjugated in-house using the Maxpar X8 Antibody Labeling Kit (Standard BioTools) following manufacturer recommendations. Platinum (Pt195/196) conjugation used isotopically purified cisplatin Pt195 or Pt196 to label partially reduced antibodies.

FFPE liver tissue sections were sectioned at 5µm onto Superfrost Plus adhesion slides and dried. Subsequently, slides were dewaxed in xylene and rehydrated through an alcohol gradient to Ultrapure water (Gibco). Slides were then subjected to pH8.5 Tris-EDTA antigen retrieval (30mins, 96°C), washed in fresh PBS, blocked in 3% BSA/PBS (45mins, RT) then stained with primary antibody mix (See S2 Table) in 0.5% BSA/PBS (overnight, 4°C) in a dark humidified chamber. Slides were then washed twice with 0.1% Triton-X/PBS (8mins, RT) and incubated with metal-tagged anti-fluorophore secondary antibodies in 0.5% BSA/PBS (2h, RT). Slides were washed twice (as before) then incubated with Ir191/193 DNA intercalator (Standard BioTools) 1:400 in PBS (30mins, RT). Slides were then rinsed once with PBS, once with Ultrapure water (Gibco) and finally air-dried.

Regions of interest were selected by histopathological features using whole-slide brightfield imaging of H&E-stained sequential sections, defined in CyTOF software (Standard BioTools) and acquired using the Hyperion Imaging Mass Cytometry system (Standard BioTools) calibrated following manufacturer directions. Acquired images were inspected and exported using MCDViewer (Standard BioTools) and analysed using ImageJ.

## Western blot

Protein was extracted from liver tissue by bead mill homogenisation in RIPA buffer. Homogenates were centrifuged for 15 minutes at 13000g at 4 degrees and the supernatant taken forward for analysis. Protein concentrations were determined using a BCA assay and normalised between samples. A standard chemiluminescence-based immunoblot protocol was followed to quantify expression levels of protein relative to total protein transfer on the blot as determined by Stain Free Imaging on a Bio-Rad ChemiDoc. Anti-Collagen 1 primary antibody (Southern Biotech, 1:500) and an anti-goat HRP secondary antibody (GE Healthcare, 1:10000) were used.

## Supporting information

**S1 Fig. Lobe-level view of fibrotic granuloma formation during infection timecourse and comparison of granuloma and peri-granuloma levels of SOX9.**
(DOCX)

**S2 Fig. Experimental time course and survival curve.**
(DOCX)

**S3 Fig. Representative Lobe level view of PSR and aSMA staining.**
(DOCX)

**S4 Fig. Quantification of necrosis.**
(DOCX)

**S5 Fig. Immune cell distribution in infected control and SOX9 KO animals.**
(DOCX)

**S6 Fig. Extended Hyperion data shows fragmented ECM deposition in Sox9 deficient animals.**
(DOCX)

**S7 Fig. Additional immune characterisation.**
(DOCX)

**S1 Table. Antibodies used for Flow Cytometry.**
(DOCX)

**S2 Table. Antibodies used for Hyperion image acquisition.**
(DOCX)

**S1 Data Raw data supporting figures and tables.**
(XLSX)

## Acknowledgments

The University of Manchester Biological Services Facility is acknowledged for their assistance running *in vivo* models.

## Author contributions

**Conceptualization:** Andrew S MacDonald, Karen Piper Hanley.

**Data curation:** Karen Piper Hanley.

**Formal analysis:** Kim Su, Elliot Jokl, Alice Costain, Kevin N Couper, Andrew S MacDonald, Karen Piper Hanley.

**Funding acquisition:** Andrew S MacDonald, Karen Piper Hanley.

**Investigation:** Kim Su, Elliot Jokl, Alice Costain, Kara Simpson, Antonn Cheeseman, Alexander Phythian-Adams, Andrew S MacDonald, Karen Piper Hanley.

**Methodology:** Kim Su, Elliot Jokl, Alice Costain, Kevin N Couper, Karen Piper Hanley.

**Project administration:** Andrew S MacDonald, Karen Piper Hanley.

**Resources:** Karen Piper Hanley.

**Supervision:** Andrew S MacDonald, Karen Piper Hanley.

**Validation:** Karen Piper Hanley.

**Visualization:** Karen Piper Hanley.

**Writing – original draft:** Kim Su, Elliot Jokl, Alice Costain, Andrew S MacDonald, Karen Piper Hanley.

**Writing – review & editing:** Kim Su, Elliot Jokl, Alice Costain, Andrew S MacDonald, Karen Piper Hanley.

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
