## [Decision Letter · Decision Letter 0]

10 Mar 2025

PPATHOGENS-D-25-00178

SOX9 plays an essential role in myofibroblast driven hepatic granuloma integrity and parenchymal repair during schistosomiasis-induced liver damage

PLOS Pathogens

Dear Dr. Piper Hanley,

Thank you for submitting your manuscript to PLOS Pathogens. After careful consideration, we feel that it has merit but does not fully meet PLOS Pathogens's publication criteria as it currently stands. Therefore, we invite you to submit a revised version of the manuscript that addresses the points raised during the review process.

Please submit your revised manuscript within 30 days May 09 2025 11:59PM. If you will need more time than this to complete your revisions, please reply to this message or contact the journal office at plospathogens@plos.org. Please include the following items when submitting your revised manuscript:

We look forward to receiving your revised manuscript.

Kind regards,

Agostinho Carvalho

Academic Editor

PLOS Pathogens

Jeffrey Dvorin

Section Editor

PLOS Pathogens

Sumita Bhaduri-McIntosh

Editor-in-Chief

PLOS Pathogens

orcid.org/0000-0003-2946-9497

Michael Malim

Editor-in-Chief

PLOS Pathogens

orcid.org/0000-0002-7699-2064

**Journal Requirements:**

At this stage, the following Authors/Authors require contributions: Andrew MacDonald, and Karen Piper Hanley. Please ensure that the full contributions of each author are acknowledged in the "Add/Edit/Remove Authors" section of our submission form.

- ® on page: 15

- TM on pages: 14, and 15.

5) We have noticed that you have uploaded Supporting Information files, but you have not included a list of legends. Please add a full list of legends for your Supporting Information files after the references list.

6) We note that your Data Availability Statement is currently as follows: "All relevant data are within the manuscript and its Supporting Information files.". Please confirm at this time whether or not your submission contains all raw data required to replicate the results of your study. Authors must share the “minimal data set” for their submission. PLOS defines the minimal data set to consist of the data required to replicate all study findings reported in the article, as well as related metadata and methods (https://journals.plos.org/plosone/s/data-availability#loc-minimal-data-set-definition).

7) Please amend your detailed Financial Disclosure statement. This is published with the article. It must therefore be completed in full sentences and contain the exact wording you wish to be published.

8) Please ensure that the funders and grant numbers match between the Financial Disclosure field and the Funding Information tab in your submission form. Note that the funders must be provided in the same order in both places as well. Currently, " AM was supported by funding from the Lydia Becker Institute and  KS was supported by the North West MRC Fellowship in Clinical Pharmacology & Therapeutics" are missing from the Funding Information tab . 

**Reviewers' Comments:**

Reviewer's Responses to Questions

**Part I - Summary**

Reviewer #1: I would like to congratulate authors to the excellent concise work describing clearly effect of SOX9 transcription factor in hepatic granuloma formation. The studies are addressed precisely and experimental data are presented clearly. It does not overwhelm the reader with excessive amount of data, but delivers information sufficiently.

Study is solely focused on the aspect of the liver infection and does not include additional studies investigating effect in the gut. The study does not provide any additional information on the fitness of the infected animals. It would be interesting to investigate healing after PZQ treatment.

Reviewer #2: This study by Su et al examines the role of the transcription factor SOX9 in liver fibrosis during murine schistosomiasis. This follows on from the authors’ previous studies showing the importance of SOX9 in chemical-induced and other models of fibrosis, and now takes this into a distinct (and strongly type 2 infection) model. They show that SOX9 is upregulated in the livers of schistosome-infected mice and SOX9 deficiency disrupts granuloma formation with a more dispersed inflammatory infiltrate that is coincident with reduced survival – this fits well with the known role of egg granulomas in preventing more widespread hepatotoxicity. The manuscript is well written and results generally clear, but I do have some suggestions and requests for clarification.

Reviewer #3: This study by Su et al. report on the role of Sox9 in hepatic fibrosis induced by the eggs of Schistosoma mansoni, using a mouse model of infection. Figure 1 demonstrate that Sox9 is progressively expressed in several cell types in infected mouse livers. The authors then address the function by using an inducible Sox9 knockout. The data presented suggest that Sox9 play an important function in granuloma formation as the infected KO’s had significantly smaller granulomas with less collagen production, and with a more diffuse pattern of fibrosis. The cell composition was altered with a significant decrease in CD4 cells together with alterations in the myeloid compartment. Only minor changes in CD4 cytokine production were observed.

The manuscript is very well written, it is clear, concise and follows a logical progression. The data is presented well and the histology is beautiful. The experiments are well designed and well executed, with relevant controls and appropriate group sizes. The discussion is clear and focussed and does not over interpret the data. Overall, I found the manuscript a pleasure to read. The subject has a broad appeal, not only to those that work on schistosomiasis. but also to a wider audience with an interest in the development and management of fibrotic diseases in general.

I have no major concerns regarding the quality of the manuscript as it is.

As a side, it is a shame that the Sox 9 KO experiments had to be terminated at day 51 for welfare reasons, but with the increased mortality in this group at this stage that is perfectly reasonable and necessary, and it does suggest a very significant host protective effect of Sox9 in the development of the granulomas. The only thing that I would have liked to see is the inclusion of FoxP3+ cells in the analysis of liver cells in Figure 4. Although IL-10+ CD4+ cells were not altered it may be that alterations in T reg populations (via IL-10 or other regulatory mechanisms) may impact on the outcome. But this is minor gap in what is otherwise a nice well-presented set of data.

**Part II – Major Issues: Key Experiments Required for Acceptance**

Reviewer #1: none

Reviewer #2: (No Response)

Reviewer #3: None

**Part III – Minor Issues: Editorial and Data Presentation Modifications**

Reviewer #1: none

Reviewer #2: - Fig. 1. Can levels of SOX9 within granulomas vs adjacent to granulomas be quantified? Figure legend states SOX9 localises to areas of SMA and PSR stain, but it appears much of the brown SOX9 stain is adjacent to the granuloma (if delineated by PSR staining on what appear to be sequential sections). This is also be case in Fig. 2A where intense SOX9 red stain is adjacent to what might be an egg (?). It makes sense if SOX9 is adjacent to but not within the granulomas given expression by damaged hepatocytes rather than the immune infiltrate. Fig. 2B white * for egg referred to in legend is missing?

- Supp Fig. 2B. There are perhaps 2x very early deaths in the SOX9-/- mice. Is there any evidence that lung inflammatory responses to migrating larvae are exaggerated in SOX9-/- mice? The additional wave of deaths ~wks6-7 is consistent with egg-induced pathology and convincing given the granuloma changes shown. I also wondered if there were any changes in intestinal inflammation/fibrosis due to egg migration, but this is perhaps more beyond scope of study.

- Fig. 3E. Representative Western does not really back up the graphed data i.e. wide variation in (stain-free) total protein levels between different samples. Assuming this is effectively a loading control, are COL1 levels corrected for this in the graph? There are only 2x naïve and 2x infected SOX9ko samples for the Western with big difference in total protein levels within the (small) groups. Given there are 5-7 infected mouse datapoints on the graph, it would be good to have more convincing representative blots.

- Page 7, “trend towards increased necrosis in the SOX9-/- infected group, but this was not significant”. Given p=0.36 and only 2/7 mice show elevated necrosis, this is a bit of a stretch. I suggest either repeating infection experiment or removing this statement.

- Were circulating liver enzymes (e.g. AST/ALT) measured? This would support increased liver damage in KO mice.

- Fig. 4. Do you have liver cell number data (total cell counts and for individual cell populations)? This would answer whether eosinophils are genuinely elevated in the KO mice, or whether another cell type (macrophages?) are reduced, and so eosinophils become proportionally greater. Does the increase in CD25(+) T cells reflect stronger T cell activation (potentially not, given ICC data) or more Tregs? This could be answered with Foxp3 staining.

- What is the justification for presenting cytokine levels relative to respective WT or KO control groups? Were there any differences between WT and KO naïve levels? And was this because the results are pooled from multiple experiments with different cytokine levels across harvests?

- Was ST2 and intracellular cytokine staining carried out on the same cells? It would be good to correlate e.g. type 2 cytokines with IL-33 receptor expression if possible.

- Supp Table 1 gives details on several Ab that are not included in the study (including Foxp3 as mentioned above). Do you have data on RELMα expression in liver macrophages and eosinophils? It would be good to see how this compares to liver Ym-1 (with dual stain for F4/80 or other macrophage markers). This is also interesting given elevated % eosinophils in KO liver (are they more activated as well as more prevalent?). Similarly, iNOS staining was carried out according to Supp Table so does this show if KO liver macrophages adopt any M1 bias? If data is not included/was not carried out then please amend table.

- Supp Fig. 5B. I am not sure if I am misreading but is the data in correct order? i.e. top right naïve KO appears to have an egg surrounded by Ym-1(+) cells, so this could be WT infected? In addition, it would be good to combine Ym1 stain with F4/80 to determine if macrophages, as per point above (i.e. are the Ym1+ cells macrophages?)

- Page 13 M/M indicates that egg counts in liver and intestine were performed. Please include this data.

- Fig. 5 Hyperion data is very impressive. Is there any data to identify the proliferating (Ki67+) cells that are mostly on the edges of the granuloma in WT mice? Additionally, can proliferating cells be quantified to show whether their numbers as well as spatial organisation changes in KO mice?

- Page 11, “infected SOX9-/- mice display an enhanced potential to produce the immunoregulatory cytokine IL-10” is not case in Fig. 4D.

- Details on intracellular cytokine staining (PMA, ionomycin, brefeldin etc) missing from M/M.

Reviewer #3: None

PLOS authors have the option to publish the peer review history of their article (what does this mean? ). If published, this will include your full peer review and any attached files.

**Do you want your identity to be public for this peer review?** For information about this choice, including consent withdrawal, please see our Privacy Policy .

Reviewer #1: No

Reviewer #2: No

Reviewer #3: No

**Figure resubmission:**
---

## [Decision Letter · Decision Letter 1]

29 Apr 2025

Dear Dr Piper Hanley,

We are pleased to inform you that your manuscript 'SOX9 plays an essential role in myofibroblast driven hepatic granuloma integrity and parenchymal repair during schistosomiasis-induced liver damage' has been provisionally accepted for publication in PLOS Pathogens.

Best regards,

Agostinho Carvalho

Academic Editor

PLOS Pathogens

Jeffrey Dvorin

Section Editor

PLOS Pathogens

Sumita Bhaduri-McIntosh

Editor-in-Chief

PLOS Pathogens

orcid.org/0000-0003-2946-9497

Michael Malim

Editor-in-Chief

PLOS Pathogens

orcid.org/0000-0002-7699-2064

Reviewer Comments (if any, and for reference):

Reviewer's Responses to Questions

**Part I - Summary**

Reviewer #1: The article is the shape to be published in PLoS Pathogens.

Reviewer #2: Happy with changes, I like the paper!

Reviewer #3: The authors have improved the manuscript sufficiently and I have no further comments

**Part II – Major Issues: Key Experiments Required for Acceptance**

Reviewer #1: no

Reviewer #2: (No Response)

Reviewer #3: N/A

**Part III – Minor Issues: Editorial and Data Presentation Modifications**

Reviewer #1: no

Reviewer #2: (No Response)

Reviewer #3: N/A

PLOS authors have the option to publish the peer review history of their article (what does this mean? ). If published, this will include your full peer review and any attached files.

**Do you want your identity to be public for this peer review?** For information about this choice, including consent withdrawal, please see our Privacy Policy .

Reviewer #1: No

Reviewer #2: No

Reviewer #3: No

---

## [Editor Report · Acceptance letter]

Dear Dr Piper Hanley,

We are delighted to inform you that your manuscript, "SOX9 plays an essential role in myofibroblast driven hepatic granuloma integrity and parenchymal repair during schistosomiasis-induced liver damage," has been formally accepted for publication in PLOS Pathogens.

Best regards,

Sumita Bhaduri-McIntosh

Editor-in-Chief

PLOS Pathogens

orcid.org/0000-0003-2946-9497

Michael Malim

Editor-in-Chief

PLOS Pathogens

orcid.org/0000-0002-7699-2064